# Changes and predictors of premarital sex intercourse among never-married women (15–24 years) in Nigeria: A multilevel approach

Joseph Ayodeji Kupoluyi ᵢD*

Department of Demography and Social Statistics, Obafemi Awolowo University, Ile-Ife, Nigeria

* jakupoluyi@gmail.com

## Abstract

### Introduction

Premarital sexual intercourse (PSI) predisposes never-married women to unwanted pregnancy, abortion, sexually transmitted infections, HIV/AIDS, and high rates of school dropout. This study examines changes and predictors of PSI among never-married women (15–24) in Nigeria.

### Methods

The study used a 15-year duration of three wave's pooled individual recode (IR) dataset of the Nigeria Demographic and Health Surveys (NDHS) of 2008, 2013, and 2018. A weighted sample size of 23,446 never-married young women was analysed using frequency tables, charts, and multilevel binary logistic regression.

### Results

The prevalence of PSI among never-married women (15–24) over the 15-year duration of 3 waves of NDHS datasets was 31.3% [95%CI: 30.3, 32.3] in Nigeria. Prevalence of PSI steadily decreased from 36.8% in 2008 to 31.7% in 2013 and 26.6% in 2018. A 5-year percentage change between 2008 and 2013 showed that PSI declined by 13.2%, 16.2% between 2013 and 2018, and 27.7% over 10 years (2008–2018). Never-married women aged 20–24 had higher odds (OR= 7.8, 95%CI [6.9, 8.8], p < 0.001) of engaging in PSI than those aged 15–19. The community's knowledge of modern methods, literacy level, socio-economic status, education, religion, region of residence, employment status, and exposure to mass media were strongly associated with PSI. The intra-class correlation value of 10.1% indicates that contextual factors significantly explained the variations in PSI between clusters. The proportional change in variance of 41.3% explained the variability in the odds of PSI explained by each model with more terms.

**Data availability statement:** Data is available from The Nigeria Demographic and Health Surveys (NDHS) datasets (2008, 2013, and 2018) and are freely available at https://dhspro-gram.com/data/available-datasets.cfm.

**Funding:** The author(s) received no specific funding for this work.

**Competing interests:** The authors have declared that no competing interests exist.

**Abbreviations:** AIC, Akaike information criterion; AOR, adjusted odds ratio; BIC, Bayesian information criterion; CI, confidence interval; ICC, intraclass correlation coefficient; DHS, demographic and health survey; NDHS, Nigeria demographic ansd health survey; PCV, proportional change in variance; PSI, premarital sexual intercourse; SD, standard deviation; SE, standard error.

## Conclusion

This study established a steady decrease in the prevalence of PSI among never-married women (15–24) over the 15-year duration of 3 waves of NDHS datasets (2008, 2013, and 2018). Thus, we conclude that concerted efforts are required to empower young women to contribute to a further reduction in PSI to improve the general national health status of women and to ensure progress towards achieving a reduction in early pregnancies among unmarried women in Nigeria.

## Background

Premarital sexual intercourse (PSI), the sexual intercourse between unmarried persons of the opposite sex, has become a major public health concern worldwide owing to the high rates of school dropout, unwanted pregnancy, abortion, sexually transmitted infections(STIs), and HIV/AIDS among other consequences [1–4]. Across all cultures and religions, marriage is the most socially recognised, appropriate, and legitimate setting for sexual intercourse [5,6]. Although the cultural norms on marriage and sex vary from culture to culture in sub-Saharan Africa including Nigeria, PSI remains unacceptable to most cultures and religions. Thus, the increase in PSI has generated many concerns among stakeholders, researchers, and policymakers. Globally, available information has shown that the prevalence of premarital sex among young women aged 15–24 has continued to increase rapidly [7–11]. In sub-Saharan African countries for instance, the prevalence of PSI among young women aged 15–24 in 2018 was 39.4%, with country-specific prevalence varying from 5% in Comoros to 75.3% in Liberia [11]. In the same year, the prevalence of PSI in Nigeria was 38.1% [11].

Despite this high prevalence of PSI across all states or regions in Nigeria, existing literature has shown a large number of young women (15–24 years) who engaged in penetrative vaginal intercourse before marriage do not use contraception [12,13]. This situation is exacerbated by a significant increase in the total number of young people aged 15–24 years [14,15] and a declining median age at sexual debut [7–18]. In Nigeria, the median age at sexual debut was 15 years [18]. Presently, young people aged 15–24 years constitute 18.52% of its estimated population [19–21]. In addition to this, about 55.89% of young people 15–24 years were young women [20,22].

Furthermore, extant literature has documented that the majority of young women aged 15–24 in Nigeria engaged in PSI [20,23]. However, a significant number of them do not have adequate knowledge of how to prevent unwanted or unintentional pregnancy [11,24–26]. As part of the government's efforts to address high PSI in Nigeria over the last two decades, several policies programmes, and interventions in sexual and reproductive health have been put in place for young women and adolescents. The national sexual and reproductive health policies have been anchored within the framework of the national health policy. The policy recognizes the implementation of reproductive health within the context of primary health care, to ensure availability and access to complete sexual and reproductive health information and quality of services. The policy largely discouraged PSI with a strong but not openly promoting abstinence until marriage as ideal behaviour. Under the policy, PSI among young women aged 15–24 is considered illegal. However, the policy provides comprehensive sex education and contraceptive services to young people to help them make informed decisions about sexual health, including preventing unwanted pregnancies and sexually transmitted infections (STIs). In addition, the policy accentuates the need for access to reproductive health services for those who are sexually active but do not aggressively criminalise premarital sex. The social and

cultural norms around the policy vary depending on region and religious beliefs. Thus, some of these policies and programmes have been largely ineffective. In examining the factors influencing PSI, this study draws from the theory of planned behaviour and social cognitive theory. The theories emphasized the role of individual attitudes in considering whether PSI is morally acceptable or not; feeling of engaging in PSI or not (social norms), access to contraception and personal confidence (perceived control), and personal beliefs in influencing decisions about PSI, while considering factors such as family background, peer pressure, religious values, and personal commitment to a partner.

Previous studies on premarital sexual intercourse have been mainly focused on the prevalence and determinants of adolescents PSI [1,2,7,8,11,17,26,27], youth sexual debut [28], students sexual practices [8], attitude towards unwanted pregnancy [29], and risky sexual behaviour among young women [30] amongst others. Most of these studies on PSI have documented a range of individual-level factors such as; respondent's age, region of residence, place of residence, wealth status, employment status, religion, and education among others as predictors of PSI [1,2,7–12,17–19,26,27]. To the best of our knowledge, there is a paucity of information on studies on community-level factors on PSI in Nigeria. Community-level factors refer to different characteristics of the social group to which individuals belong. These characteristics may affect PSI either directly or indirectly by controlling how individuals' characteristics affect PSI. It is therefore important to know the effect of community-level variables on individual/household-level outcomes on PSI. The availability of DHS hierarchical data in which individuals are clustered within households and households clustered within the communities, gives room to account for the effect of both the individual and group-level influence on PSI in Nigeria. In addition to this, there is a dearth of studies on the general change in premarital sexual intercourse among young women using longitudinal nationally representative data, to examine the progress made in reducing PSI in Nigeria. Thus, this study will fill this knowledge gap by using pooled nationally representative data and a multilevel binary logistic regression model to track PSI trends and their predictors over time in Nigeria. The use of 15-year duration of 3- waves NDHS data offers new insights that previous studies could not. Thus, this study aimed to investigate changes and predictors of premarital sexual intercourse among young women in Nigeria. Assessing the changes and predictors of PSI could inform the development of policies and programmes on PSI among young women in Nigeria that will empower young women to make informed choices about their sexual and reproductive health rights.

## Methods

### Data source

The Nigeria Demographic and Health Survey (NDHS) women's recode (IR) dataset of 2008, 2013, and 2018 was pooled and used for this study. The 15-year duration of 3 waves pooled NDHS IR files were datasets for women of reproductive age of 15 and 49 years. The datasets were pooled to increase the sample size or number of observations and to enable studying changes in PSI over time. It also enhanced statistical power, the ability to compare PSI and validate models across settings, and opportunities to develop new measures. The NDHS is a nationally representative survey that collects information on socio-demographic characteristics and health-related issues. The NDHS data of 2008, 2013, and 2018 are the fourth, fifth, and sixth rounds of DHS surveys in Nigeria. The dataset can be accessed using the URL: https://www.dhsprogram.com/data/available-datasets.cfm. A pooled sample size of 23,446 never-married women comprising 6,940, 7,744, and 8,763 never-married women for 2008, 2013, and 2018 respectively were included in this study.

## Study design

The study used three waves of repeated cross-sectional surveys conducted over a 15-year duration. The Nigeria Demographic and Health Survey (NDHS) were conducted in 2008, 2013, and 2018 using a cross-sectional study design. DHS used a two-stage sampling procedure to collect data from all 36 states and the federal capital territory (FCT), Abuja, Nigeria. The samples for the survey were drawn randomly from cluster or enumeration areas (EAs). The EAs served as the primary sampling unit for the survey. A systematic selection of households from the list of households was done to interview eligible respondents. Women aged 15–49 and men aged 15–64 were selected systematically from the households and interviewed. A detailed explanation of the sample design of the DHS surveys has been published previously [13,31,32] and is freely available at https://www.dhsprogram.com.

## Measurement of variables

**Outcome variable.** Premarital sexual intercourse (PSI) was the outcome variable of this study. It was derived and operationalised by asking respondents their age at sexual debut before marriage. That is, "At what age did the respondent first have sex?" Women who acknowledged never married and had had sex at a minimum age of 8 (the DHS defined minimum age at first sexual intercourse), were classified as "had PSI" and were coded as "1" while never-married women who had never had sex were classified as "No PSI" and coded as "0". Other categories of responses in the variable (97, 98, and 99) were treated as "missing values" and were dropped from the analysis.

**Explanatory variables.** All the explanatory variables used for this study were selected based on the established association reported in the previous studies with PSI and their level of significance ($p < 0.05$). A Multicollinearity test was performed using a variance inflation factor (VIF < 5). All explanatory variables with evidence of no collinearity (mean VIF= 1.38, maximum = 1.87 and minimum VIF = 1.10) were retained in the models while those with evidence of collinearity (VIF > 5) were excluded from the models. Explanatory variables were grouped into individual-level (socio-demographic and economic) and contextual factors. The individual level variables were education (no formal education, primary, secondary, tertiary); wealth quintile (poorest, poorer, middle, richer, richest); religious affiliation (Christian, Islam, and traditional/others); occupation (not working, working); and media exposure which was constructed by aggregating exposure to media. Combined frequencies of reading newspapers, watching television and listening to radio within a week. The scores were further distributed into two groups (exposed and not exposed). DHS measured the wealth index as a uniform composite variable determined through principal component analysis (PCA) and was built on household properties such as water supply, television, electricity, radio, refrigerator, type of flooring, and type of vehicle. The scores were categorised into five quintiles, and each quintile represents a relative measure of a household's socioeconomic status Contextual variables considered in this study were community socioeconomic status, community knowledge of modern contraceptives, community literacy and region of residence. The region of residence was measured and divided into five geopolitical groups namely: South West, South East, North West, North East, North Central, and South-South. Other contextual variables were constructed and classified by aggregating individual-level factors at the cluster level using the median value as a cut-off point. These variables were: community socioeconomic status (low, moderate, high), community literacy (low, moderate, high), and community knowledge of modern contraceptives

(low, moderate, high). The community socioeconomic status was generated from the combination of individual wealth quintiles while community literacy was generated from the combination of individual literacy levels. Community knowledge of modern contraceptives was generated from the combination of individual knowledge of any contraceptive methods.

## Statistical analyses

The objectives of the study are to observe changes and the determinants of PSI among never-married women (15–24) in Nigeria. In achieving this, a 15-year duration pooled NDHS dataset (IR dataset of NDHS, 2008, 2013, and 2018) was used. Data were analysed using Stata version 14. The weights and svy command in Stata were declared and applied to adjust for the over/under-sampling and nonresponse during the survey. Descriptive statistics (frequencies, percentages table, and bar chart) were performed to observe the prevalence and changes in PSI. Mann Kendall test was conducted to detect trends in time. It is a test that helps us know whether a trend exists at a time and whether it is statistically significant. To identify potential covariates and measure the strength of the association between explanatory variables and the outcome variable, inferential statistics (Cross-tabulation and Pearson chi-square test of independence at the 95% level of significance, and multilevel mixed effect binary logistic regression) was employed at the bivariable and multivariable levels respectively. Also, a sensitivity test was conducted to show the relationship between age and other explanatory variables. A multicollinearity test was also performed among the explanatory variables using a variance inflation factor (VIF) < 5 as the fixed point. All explanatory variables with no significant evidence of collinearity (mean VIF= 1.38, maximum = 1.87 and minimum VIF = 1.10) were retained in the analysis. Finally, owing to the hierarchical nature of the NDHS data and the objective of this study, a four-model multi-level binary logistic regression was constructed to examine the individual and community-level factors associated with PSI. Model 0, the empty model had no explanatory variable, Model 1 had only the individual variables, Model II had only the contextual variables and lastly, Model III had both individual and contextual/community variables. Adjusted odds ratio (AOR) and 95% confidence intervals (CI) were provided for all models in the multivariable analysis. The Intra-Class Correlation (ICC) and the Proportional Change in Variance (PVC) were calculated to measure the random effects within or between clusters.

The ICC was defined as $ICC = \frac{VA}{\left(VA + 3.29\right)} \times 100$ Where VA represents the estimated variance.

The PCV was also calculated as $PCV = \frac{VA - VB}{VA} \times 100$ Where VA is the variance in PSI in the empty model and VB is the variance in successive models. The goodness of fit of the regression models was determined using Akaike information criteria (AIC). A lower value of AIC indicates a better fit [33].

## Handling of missing values

Missing values on age at first sex (i.e., missing and unknown) are considered as not having had sexual intercourse. Thus, they are excluded from the analysis. Furthermore, missing values in other variables were declared missing and therefore, removed from the analysis. In addition, to make the sample representative of the population as recommended by DHS, sample weights and svy command in Stata were declared and applied to adjust for the over/under-sampling and nonresponse during the survey.

### Ethical consideration

In this study, permission for the usage of all the NDHS datasets was sought and granted by Measuredhs. Being secondary data, details of other ethical considerations are available free at http://googl/ny8T6X.

## Results

### Background characteristics of the respondents

The percentage distribution of the respondents' socio-demographic characteristics is presented in Table 1. The pooled result showed that the majority of the respondents (70.7%) were in the age group of 15–19 years with a mean age (± Standard Deviation [SD]) of 18.1 (±2.7) years. The proportion of respondents in the same age group (15–19) increased across the survey year from 66.1% in 2008 to 71.1% in 2013 and 73.9% in 2018. Across the survey year, the majority of the respondents have had secondary education though the proportion decreased from 75.1% in 2008 to 70.9% in 2018. Also, between 2008 and 2018, more than four in five respondents (82.2%) had secondary or tertiary education. There are more Christians (65%) in the pooled data than Islam and other religious affiliations. In the 2008, 2013, and 2018 surveys, Islam increased from 22.5% in 2008 to 44.8% in 2018. This rising pattern in the proportion of respondents across the survey year was more noticeable among those who were working, and exposed to the media, and those from the Northern part of the country, with low community literacy levels, and low community socio-economic status.

Further analysis of the percentage distribution of the respondents' socio-demographic characteristics between 2008 and 2018 shows that two-thirds (67.3%) were not working, about 30.4% were in the richest wealth status, and about 63.6% were not exposed to mass media. At least one in five respondents (21.9%) lived in the southwestern region of the country. While more than a third (36.2%) were from communities with high literacy levels, about 32.4% and 36.6% of the respondents were from communities with medium socio-economic status and communities with high knowledge of modern contraceptives respectively.

### Trends and changes in premarital sexual intercourse among young women (15–24) in Nigeria (2008–2018)

As shown in Fig 1, the prevalence of premarital sexual intercourse (PSI) among never-married women (15–24) in Nigeria showed a consistent drop in the prevalence of PSI in Nigeria (p < 0.001). PSI dropped from 36.76% [34.9–38.6] in 2008 to 31.71% [29.8–33.7] in 2013 and to 26.58% [25.1–28.1] in 2018. Overall across the three surveys, the prevalence of PSI over ten years was 31.28% [30.3–32.3] in Nigeria. Further analysis of the pattern of changes in PSI showed a 5-year percentage change in PSI between 2008 and 2013 declined by 13.18%. Also, while comparing the 5-year percentage change in PSI between 2013 and 2018, PSI decreased by 16.18%. The pattern of changes in PSI over a 15-year duration (2008, 2013 and 2018) also recorded a 27.69% decrease in PSI.

Table 2 also presents the bivariate distribution of PSI among never-married women (15–24) by selected covariates. The results show that all the explanatory variables - age, educational level, religion, employment status, wealth status, media exposure, region, community literacy level, community knowledge of modern contraceptives, and community socioeconomic status -were significantly associated with PSI at p <.001 except community socio-economic status across the survey year. The prevalence of PSI was statistically significantly higher in women who have had tertiary education (p= 53.6% [50.5, 56,7], p<0.001), are aged 20–24 (p = 60.19% [58.4, 62.0], p < 0.001), are Christians (p = 40.95 [39.8, 42.1], p < 0.001), working (p = 43.15

**Table 1. Percentage distribution of the socio-demographic characteristics of never-married women (15-24) according to the survey year.**

| Variable | 2008 | 2013 | 2018 | 2008-2018 |
|---|---|---|---|---|
| | N (%) | N (%) | N (%) | N (%) |
| Individual-level factors | | | | |
| Age | | | | |
| 15-19 | 4586 (66.1) | 5508 (71.1) | 6471 (73.8) | 16565 (70.6) |
| 20-24 | 2354 (33.9) | 2236 (28.9) | 2292 (26.2) | 6881 (29.4) |
| Educational level | | | | |
| No formal education | 319 (4.6) | 639 (8.3) | 951 (10.9) | 1909 (8.1) |
| Primary | 803 (11.6) | 737 (9.5) | 731 (8.3) | 2271 (9.7) |
| Secondary | 5208 (75.1) | 5697 (73.6) | 6215 (70.9) | 17121 (73.0) |
| Tertiary | 610 (8.9) | 671 (8.6) | 866 (9.9) | 2146 (9.2) |
| Religion | | | | |
| Christian | 5323 (76.9) | 5077 (65.8) | 4791 (54.7) | 15192 (64.9) |
| Islam | 1554 (22.5) | 2579 (33.4) | 3928 (44.8) | 8061 (34.5) |
| Others | 38 (0.6) | 57 (0.7) | 44 (0.5) | 139 (0.6) |
| Employment status | | | | |
| Not working | 5002 (72.8) | 5446 (71.0) | 5227 (59.6) | 15675 (67.3) |
| Working | 1868 (27.2) | 2228 (29.0) | 3536 (40.4) | 7631 (32.7) |
| Wealth status | | | | |
| Poorest | 520 (7.5) | 514 (6.6) | 895 (10.2) | 1929 (8.2) |
| Poorer | 849 (12.2) | 1002 (12.9) | 1296 (14.8) | 3147 (13.4) |
| Middle | 1412 (20.3) | 1739 (22.5) | 1762 (20.1) | 4913 (21.0) |
| Richer | 1930 (27.8) | 2073 (26.8) | 2317 (26.4) | 6320 (21.0) |
| Richest | 2228 (32.1) | 2416 (31.2) | 2493 (28.4) | 7137 (30.4) |
| Media exposure | | | | |
| Not Exposed | 3945 (56.9) | 4877 (63.0) | 6095 (69.6) | 14918 (63.6) |
| Exposed | 2994 (43.2) | 2867 (37.0) | 2668 (30.4) | 8529 (36.4) |
| Community-level factors | | | | |
| Region | | | | |
| North Central | 1051 (15.2) | 1247 (16.1) | 1314 (15.0) | 3612 (15.4) |
| North East | 484 (7.0) | 754 (9.7) | 1296 (14.8) | 2534 (10.2) |
| North West | 568 (8.2) | 1365 (17.6) | 1929 (22.0) | 3862 (16.5) |
| South East | 1306 (18.8) | 1369 (17.7) | 1258 (14.4) | 3932 (16.8) |
| South South | 1689 (24.3) | 1467 (18.9) | 1224 (14.0) | 4381 (18.7) |
| Southwest | 1842 (26.5) | 1542 (19.9) | 1741 (19.9) | 5124 (21.9) |
| Community literacy level | | | | |
| Low | 1995 (28.8) | 2680 (34.2) | 2590 (29.6) | 7245 (30.9) |
| Medium | 4944 (71.2) | 5093 (65.8) | 3158 (36.0) | 7714 (32.9) |
| High | 0 (0.00) | 0 (0.00) | 3014 (34.4) | 8487 (36.2) |
| Community knowledge of modern contraceptives | | | | |
| Low | 1943 (28.0) | 2694 (34.8) | 2569 (29.3) | 7064 (30.1) |
| Medium | 2305 (33.2) | 5050 (65.2) | 6194 (70.7) | 7800 (33.3) |
| High | 2692 (38.8) | 0 (0.0) | 0 (0.00) | 8582 (36.6) |
| Community socio-economic status | | | | |
| Low | 2529 (36.5) | 2936 (37.9) | 3859 (44.0) | 6935 (29.6) |
| Medium | 1588 (22.9) | 1915 (24.7) | 1401 (16.0) | 7591 (32.4) |
| High | 2822 (40.7) | 2892 (37.4) | 3502 (40.0) | 8919 (30.0) |
| **Total** | **6940 (100)** | **7744 (100)** | **8763 (100)** | **23446 (100)** |

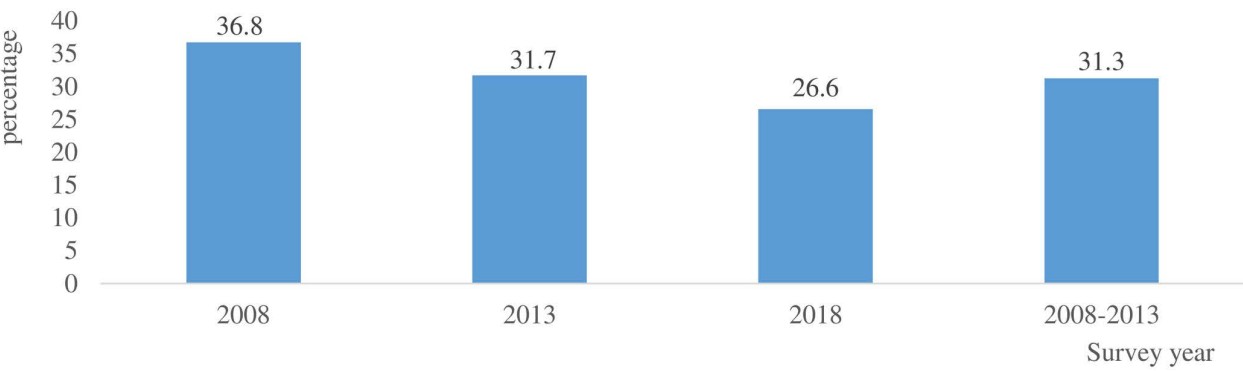

**Fig 1. Prevalence of premarital sexual intercourse among yound women in Nigeria.**

[41.5, 44.8], p < 0.001), richer (p = 35.19 [33.4, 37.1], p < 0.001), are media exposed (p = 39.61 [37.9, 41.3], p<0.001) are from the south-south region (p = 52.39 [50.3, 54.5], p < 0.001), have high community literacy level (p = 35.23 [33.3, 37.2], p < 0.001), have high community knowledge of modern contraceptives (p = 38.70 [36.9, 40.6], p < 0.001) and high community socio-economic status (p = 34.10 [32.2, 36.1], p < 0.001) than their counterparts in other categories.

## Predictors of PSI

**Fixed effects.** Table 3 showed significant variations in never-married women (15–24) engaging in PSI at both individual and community levels. Model I shows the effects of individual-level factors on PSI. The findings show that never-married women aged 20–24 (AOR = 7.8, 95%CI [6.9, 8.9], p <0.001), with a tertiary level of education (AOR = 3.3, 95%CI [2.5, 4.4], p <0.001), employed (AOR = 2.3, 95%CI [2.1, 2.5], p <0.001), exposed to media (AOR = 1.5, 95%CI [1.4, 1.6], p <0.001), and were richer (AOR = 1.3, 95%CI [1.1, 1.8}, p <0.05), wealth quintile had higher odds of engaging PSI than their counterparts who were not. There are little variations by wealth quintile. Also, never-married women who are Muslims have 20% (AOR = 0.8, 95%CI [0.5, 1.3], p <0.001) lower odds than their counterparts who are Christians. Model II shows the effects of the community-level factors on PSI. The finding shows that the south-south region of residence (AOR = 2.8, 95%CI [2.4, 3.2], p <0.001), medium community literacy level (AOR = 1.1, 95%CI [1.0, 1.3], p 0.001), high community knowledge of modern contraceptives (AOR = 1.7, 95%CI [1.5, 2.0], p <0.001), and medium community socioeconomic status (AOR = 0.7, 95%CI [0.6, 0.9], p <0.001) were significantly associated with PSI. The final (Model III) result shows the effects of both individual and community-level factors. The fixed effect of the multilevel logistics regression in Model III showed that never-married women aged 20–24 had higher odds (OR= 7.8, 95%CI [6.9, 8.8], p < 0.001) of engaging in PSI than those aged 15–19. The result shows that the odds of engaging in PSI increase significantly with the level of education. For instance, never-married women (15–24) with tertiary level of education had higher odds (OR= 2.7, 95%CI [2.0, 3.5], p < 0.001) of engaging in PSI than those with no formal education. As regards religious affiliations, never-married Islamic women had lower odds (OR= 0.4, 95%CI [0.3, 0.4], p < 0.001) of engaging in PSI than never-married Christian women. Also, never-married employed women had higher odds (OR= 2.3, 95%CI [2.1, 2.5], p < 0.001) of engaging in PSI than never-married women who were not employed. On the wealth index, the result showed that the richest never-married women had 40% lower odds (OR= 0.6, 95%CI [0.5, 0.7], p < 0.001) of engaging

**Table 2. Bivariate distribution of PSI by socio-demographic characteristics of never-married women (15-24) according to the survey year in Nigeria.**

| Variable | Premarital Sexual Intercourse | | | |
|---|---|---|---|---|
| | **2008** | **2013** | **2018** | **2008-2018** |
| | % [95%CI] | % [95%CI] | % [95%CI] | % [95%CI] |
| | (36.8) [34.9, 38.6] | (31.7) [29.8, 33.7] | (26.6) [25.1, 28.1] | (31.3) [30.3, 32.3] |
| **Individual-level factors** | | | | |
| **Age groups** | p < 0.001 | p < 0.001 | p < 0.001 | p < 0.001 |
| 15-19 | 23.6 [21.9, 25.3] | 20.09 [18.6, 21.6] | 15.6 [14.5, 16.8] | 19.31 [18.4, 20.2] |
| 20-24 | 62.5 [59.9, 65.1] | 60.48 [57.4, 63.4] | 57.5 [54.5, 60.5] | 60.19 [58.4, 62.0] |
| $\chi2$ | 978.4 | 1258.17 | 1541 | 3819.3 |
| **Educational level** | p < 0.001 | p < 0.001 | p < 0.001 | p < 0.001 |
| No formal education | 10.1 [7.1, 14.3] | 7.6 [5.5, 10.3] | 8.8 [7.0, 11.1] | 8.61 [7.24, 10.2] |
| Primary | 28.4 [24.6, 32.6] | 26.4 [22.3, 30.8] | 21.4 [18.0, 25.2] | 25.47 [23.2, 27.9] |
| Secondary | 36.8 [34.8, 38.8] | 32.0 [30.2, 33.9] | 27.4 [25.8, 29.0] | 31.77 [30.6, 32.9] |
| Tertiary | 61.6 [56.4, 66.6] | 57.8 [52.2, 63.3] | 44.7 [40.2, 49.3] | 53.60 [50.5, 56,7] |
| $\chi2$ | 273.6 | 414.2 | 316.1 | 1003.7 |
| **Religion** | p < 0.001 | p < 0.001 | p < 0.001 | p < 0.001 |
| Christian | 41.7 [39.7, 43.7] | 41.7 [39.8, 43.6] | 39.4 [37.7, 41.1] | 40.95 [39.8, 42.1] |
| Islam | 20.0 [17.1, 23.4] | 12.2 [10.3, 14.3] | 11.0 [9.0, 13.5] | 13.13 [11.7, 14.7] |
| Others | 35.1 [20.8, 52.7] | 27.2 [10.2, 54.9] | 24.6 [13.5, 40.6] | 28.51 [18.6, 41.1] |
| $\chi2$ | 235.4 | 725.4 | 898.9 | 1922.8 |
| **Employment status** | p < 0.001 | p < 0.001 | p < 0.001 | p < 0.001 |
| Not working | 31.3 [29.5, 33.2] | 26.7 [24.9, 28.6] | 18.8 [17.4, 20.3] | 25.51 [24.4, 26.6] |
| Working | 51.5 [48.4, 54.7] | 44.2 [41.3, 47.1] | 38.1 [36.0, 40.3] | 43.15 [41.5, 44.8] |
| $\chi2$ | 229.9 | 235.7 | 408.9 | 751.2 |
| **Wealth status** | p < 0.001 | p < 0.001 | p < 0.001 | p < 0.001 |
| Poorest | 32.3 [27.1, 38.0] | 11.64 [8.52, 15.7] | 15.8 [12.5, 19.9] | 19.15 [16.6, 21.9] |
| Poorer | 33.1 [29.3, 37.1] | 27.27 [23.7, 31.2] | 23.7 [20.7, 27.0] | 27.36 [25.3, 29.6] |
| Middle | 35.6 [32.6, 38.8] | 32.31 [29.3, 35.5] | 27.3 [24.8, 29.9] | 31.46 [29.7, 33.3] |
| Richer | 42.2 [38.9, 45.5] | 33.51 [30.8, 36.4] | 30.9 [28.2, 33.8] | 35.19 [33.4, 37.1] |
| Richest | 35.2 [32.2, 38.4] | 35.81 [32.9, 38.9] | 27.4 [24.8, 30.2] | 32.70 [30.9, 34.5] |
| $\chi2$ | 35.5 | 133.4 | 83.3 | 208.6 |
| **Media exposure** | p < 0.001 | p < 0.001 | p < 0.001 | p < 0.001 |
| Not Exposed | 30.3 [28.3, 32.3] | 27.1 [25.3, 28.8] | 23.7 [22.1, 25.3] | 26.52 [25.4, 27.7] |
| Exposed | 45.4 [42.8, 48.0] | 39.6 [36.9, 42.4] | 33.2 [30.2, 36.3] | 39.61 [37.9, 41.3] |
| $\chi2$ | 161.9 | 138.8 | 86 | 437.1 |
| **Community-level factors** | | | | |
| Region | p < 0.001 | p < 0.001 | p < 0.001 | p < 0.001 |
| North Central | 28.5 [25.1, 32.2] | 25.1 [22.1, 28.3] | 32.8 [30.0, 35.6] | 28.86 [27.0, 30.8] |
| North East | 21.0 [15.9, 27.3] | 15.4 [11.9, 19.6] | 15.7 [12.7, 19.1] | 16.60 [14.2, 19.3] |
| North West | 7.24 [4.2, 12.3] | 12.0 [8.0, 17.7] | 4.3 [2.7, 6.7] | 7.45 [5.3, 10.4] |
| South East | 34.6 [31.1, 38.2] | 38.5 [35.1, 42.1] | 33.3 [29.8, 36.9] | 35.53 [33.5, 37.6] |
| South South | 57.6 [54.0, 61.1] | 50.6 [47.2, 53.9] | 47.4 [43.8, 51.1] | 52.39 [50.3, 54.5] |
| Southwest | 37.2 [33.6, 40.9] | 38.5 [35.0, 42.1] | 35.2 [32.7, 37.9] | 36.91 [35.0, 38.9] |
| $\chi2$ p-value | 593.5 | 701.5 | 977.2 | 2331.6 |
| **Community literacy level** | p < 0.001 | p < 0.001 | p < 0.001 | p < 0.001 |
| Low | 29.6 [26.3, 33.1] | 22.8 [20.2, 25.7] | 21.2 [19.0, 23.6] | 26.08 [24.4, 27.9] |
| Medium | 39.7 [37.5, 41.9] | 36.3 [34.0, 38.7] | 27.5 [24.8, 30.3] | 31.82 [29.8, 33.9] |
| High | n.a | n.a | 30.2 [27.5, 33.1] | 35.23 [33.3, 37.2] |

*(Continued)*

**Table 2.** (Continued)

| Variable | Premarital Sexual Intercourse | | | |
|---|---|---|---|---|
| | 2008 | 2013 | 2018 | 2008-2018 |
| | % [95%CI] | % [95%CI] | % [95%CI] | % [95%CI] |
| | (36.8) [34.9, 38.6] | (31.7) [29.8, 33.7] | (26.6) [25.1, 28.1] | (31.3) [30.3, 32.3] |
| $\chi^2$ | 60.5 | 154.3 | 61.6 | 155.5 |
| Community knowledge of modern contraceptives | p < 0.001 | p < 0.001 | p < 0.001 | p < 0.001 |
| Low | 22.9 [20.0, 25.9] | 18.2 [15.6, 21.2] | 13.7 [12.0, 15.6] | 21.89 [20.1, 23.7] |
| Medium | 39.7 [36.6, 42.9] | 38.9 [36.8, 41.0] | 31.9 [30.1, 33.8] | 31.61 [29.8, 33.5] |
| High | 44.3 [41.1, 47.5] | n.a | n.a | 38.70 [36.9, 40.6] |
| $\chi^2$ | 228 | 364.5 | 314.2 | 516.6 |
| Community socio-economic status | p = 0.1794 | p < 0.01 | p < 0.05 | p < 0.001 |
| Low | 34.4 [31.6, 37.3] | 27.8 [25.2, 30.6] | 24.2 [22.2, 26.3] | 28.54 [26.8, 30.3] |
| Medium | 38.7 [34.9, 42.7] | 31.0 [26.7, 35.5] | 25.7 [21.6, 30.4] | 30.46 [28.5, 32.5] |
| High | 37.8 [34.6, 41.1] | 36.2 [32.8, 39.7] | 29.6 [26.9, 32.3] | 34.10 [32.2, 36.1] |
| $\chi^2$ | 9.9 | 50.9 | 28.1 | 60.4 |
| Total | 2536 | 2446 | 2327 | 7309 |

*p<0.05, **p<0.01, ***p<0.001= p-value, C.I. = Confidence Interval n.a = Not available.

in PSI than the poorest never-married women. Also, never-married women who were exposed to media (television, radio, newspaper) had higher odds (OR= 1.4, 95%CI [1.3, 1.5], p < 0.001) of engaging in PSI than never-married women who were not exposed to media. As regards region of residence, the result showed higher odds of engaging in PSI among never-married women from the South-South (OR= 2.6, 95% CI [2.2, 3.1], p < 0.001) and Southwest (OR= 1.4, 95%CI [1.2, 1.7], p < 0.001) of the country compared to never-married women from the North Central. Also, the result showed 20% lower odds of engaging in PSI among never-married women with high community level literacy (OR= 0.8, 95% CI (0.7, 1.0)], p < 0.05) compared to never-married women with low community literacy level. The result further showed higher odds of engaging in PSI among never-married women from a community with high knowledge of modern contraceptives (OR= 1.6, 95% CI [1.3, 1.9], p < 0.001) and a community with medium knowledge of modern contraceptives (OR= 1.4, 95% CI [1.2, 1.6], p < 0.001). Never-married women with high community socio-economic status had higher odds (OR= 0.7, 95% CI [0.6, 0.9], p < 0.001) of engaging in PSI compared to those with low community socio-economic status. Finally, the result showed that never-married women in the 2018 survey year had 20% lower odds (OR= 0.8, 95% CI [0.7, 0.9], p < 0.001) of engaging in PSI than those never-married women in the 2008 survey year.

**Random effects.** The random effect of the multilevel logistics regression models presented in Table 3 shows the variations between clusters (EAs). The Intraclass correlation coefficient (ICC) explained the proportion of total variance in the PSI that is attributable to the area level while the proportional change in variance (PCV) explained the variability in the odds of PSI explained by each of the models. As shown in the empty model, the ICC of 16.0% was associated with PSI which shows the variability attributed to the community level. By comparing Empty (model 0) and Model 1, the ICC of 11.8% shows significant variation explained by individual-level factors. Besides, In Model II, the ICC of 9.9% shows significant variations in the never-married women (15–24) engaging PSI due to the effect of community-level factors. The ICC of 10.1% in Model III shows significant variations in the never-married women (15–24) engaging PSI due to the effect of both individual community-level factors. It implies that the individual and community factors were important in explaining the variations

**Table 3. Multilevel binary logistic regression analysis on the predictors of PSI among young women (15-24) in Nigeria.**

| Variable | Model (Null Model) | Model I (Individual) aOR (95% CI) | Model II (Community) aOR (95% CI) | Model III (Individual and community) aOR (95% CI) |
|---|---|---|---|---|
| **Fixed effect** | | | | |
| **Individual-level factors** | | | | |
| **Age** | | | | |
| 15-19 | | RC | | RC |
| 20-24 | | 7.8 (6.9, 8.9)*** | | 7.8 (6.9, 8.8)*** |
| **Educational level** | | | | |
| No formal education | | RC | | RC |
| Primary | | 1.8 (1.4, 2.4) *** | | 1.3 (1.0, 1.7)* |
| Secondary | | 2.2 (1.8, 2.8) *** | | 1.6 (1.3, 2.0)*** |
| Tertiary | | 3.3 (2.5, 4.4) *** | | 2.7 (2.0, 3.5)*** |
| **Religion** | | | | |
| Christian | | RC | | RC |
| Islam | | 0.2 (0.2, 0.3) *** | | 0.4 (0.3, 0.4)*** |
| Others | | 0.8 (0.5, 1.3) | | 0.8 (0.5, 1.4) |
| **Employment status** | | | | |
| Not working | | RC | | RC |
| Working | | 2.3 (2.1, 2.5) *** | | 2.3 (2.1, 2.5)*** |
| **Wealth status** | | | | |
| Poorest | | RC | | RC |
| Poorer | | 1.3 (1.1, 1.6)* | | 1.1 (0.9, 1.3) |
| Middle | | 1.3 (1.1, 1.7)* | | 0.9 (0.8, 1.2) |
| Richer | | 1.3 (1.1, 1.8)* | | 0.9 (0.8, 1.1) |
| Richest | | 0.8 (0.7, 1.0)* | | 0.6 (0.5, 0.7)*** |
| **Media exposure** | | | | |
| Not Exposed | | RC | | RC |
| exposed | | 1.5 (1.4, 1.6) *** | | 1.4 (1.3, 1.5)*** |
| **Community-level factors** | | | | |
| **Region** | | | | |
| North Central | | | RC | RC |
| North East | | | 0.4 (0.4, 0.5)*** | 0.6 (0.5, 0.7)*** |
| North West | | | 0.1 (0.1, 0.1)*** | 0.2 (0.1, 0.2)*** |
| South East | | | 1.1 (0.9, 1.3) | 0.8 (0.7, 0.9)* |
| South South | | | 2.8 (2.4, 3.2)*** | 2.6 (2.2, 3.1)*** |
| South West | | | 1.3 (1.1, 1.5)*** | 1.4 (1.2, 1.7)*** |
| **Community literacy level** | | | | |
| Low | | | RC | RC |
| Medium | | | 1.1 (1.0, 1.3) | 0.9 (0.8, 1.1) |
| High | | | 1.0 (0.9, 1.2) | 0.8 (0.7, 1.0)* |
| **Community knowledge of modern contraceptives** | | | | |
| Low | | | RC | RC |
| Medium | | | 1.5 (1.3, 1.7)*** | 1.4 (1.2, 1.6)*** |
| High | | | 1.7 (1.5, 2.0)*** | 1.6 (1.3, 1.9)*** |

*(Continued)*

**Table 3.** (Continued)

| Variable | Model | Model I | Model II | Model III |
|---|---|---|---|---|
| | (Null Model) | (Individual) | (Community) | (Individual and community) |
| | | aOR (95% CI) | aOR (95% CI) | aOR (95% CI) |
| **Community socio-economic status** | | | | |
| Low | | | RC | RC |
| Medium | | | 0.9 (0.8, 1.0) | 0.9 (0.8, 1.1) |
| High | | | 0.7 (0.6, 0.8)*** | 0.7 (0.6, 0.9)*** |
| **Survey year** | | | | |
| 2008 | | | | RC |
| 2013 | | | | 0.9 (0.9, 1.1) |
| 2018 | | | | 0.8 (0.7, 0.9)*** |
| **Random effects** | | | | |
| Community-level Variance (SE) | 0.6 (0.05)*** | 0.44 (0.05)*** | 0.36 (0.04)*** | 0.37 (0.04)*** |
| VPC= ICC% | 16 | 11.8 | 9.9 | 10.1 |
| Explained Variance PCV in (%) | Reference | 30.2 | 42.9 | 41.3 |
| **Model fitness** | | | | |
| LR test | 995.5 (p< 0.001) | 524.0 (p< 0.001) | 454.6 (p< 0.001) | 404.9 (p< 0.001) |
| Log-likelihood | -14367 | -11683 | -13386 | -11241 |
| Model fit Statistics AIC | 28739.7 | 23395.4 | 26799.9 | 22537.4 |
| BIC | 28763.9 | 23516.4 | 26912.9 | 22763.3 |

RC: Reference category, aOR: adjusted odds ratio, CI: confidence interval, *p<0.05, **p<0.01, ***p<0.001.

in PSI between clusters. It showed that 10.1% of the variability in the log odds of engaging in PSI could be attributed to differences between communities and the variations in the odds of PSI between communities however remained significant ($\tau$ = 0.37 p < 0.001). Meanwhile, the between-cluster variability declined over successive models from 16% in the empty model to 9.9% in Model II then increased to 10.1% in the combined model (Model III). The PCV of 41.3% in Model III also explained the variability in the odds of PSI explained by each of the models with more terms. It indicated that 41.3% of the variation in PSI across communities was explained by individual and community-level factors. Lower values of AIC and BIC in the final model show an improvement and a better-fitted model over the previous models.

## Discussion

This study examined changes and predictors of PSI among never-married women (15–24) in Nigeria. This was done to provide more information on the sexual and reproductive health of never-married women (15–24) in Nigeria. The study found a high prevalence of PSI among never-married women (15–24) over ten years in Nigeria. This is substantiated by a recent study [11] in sub-Saharan Africa (SSA) that found a high prevalence of PSI among young women in Nigeria. In addition, the study found a consistent downward trend in the prevalence of PSI among never-married women (15–24) from 2008 to 2018. The downward trend is similar to the high prevalence of PSI (39.4%) reported among young women in sub-Saharan Africa (SSA) [11]. This downward trend could be attributed to the government and stakeholders' collective efforts in engaging the youth on sexual and reproductive health information and services over the years in the country. It could also be attributed to the underreporting of engagement in PSI among never-married women out of fear of being seen as promiscuous.

Educating never-married women should be promoted on the consequences of PSI. This study also found higher odds of engaging in PSI with an increasing age. Previous studies [2,3,7–9,11,12,17,23] have reported that young women aged 20–24 engaged in PSI than young women aged 15–19. This could be a result of the freedom and exposure older women (20–24) exercised over their younger women aged 15–19 who are likely to be in school or under the strict control of their parents. In line with the previous studies [2,3,7–9,11,12,17,23], the study found an increasing likelihood of engaging in PSI among never-married women (15–24) as their educational level increased. This could be a result of peer influences, sex and relationship talks among peers, and awareness of the use of contraception among their peers in school amongst other reasons. As regards religious affiliations, the study found lower odds of engaging in PSI among never-married Islamic women relative to other religious affiliations. Previous studies [11,13,17] have reported lower odds of premarital sex among Muslim girls compared to Christians and Hindus. It has been asserted that religion has a powerful influence on sexual behaviour with Islam being the most successful religion in putting forth religious precepts about premarital sex and getting its adherents to abide by these precepts. Other factors that might have influenced the finding include age at marriage for a Muslim girl, restriction on women's mobility and the fact that Muslims are less likely to report honestly about their transgression if they had engaged in PSI than other religious affiliations. Our findings also reveal that never-married women (15–24) who were employed had higher odds of engaging in PSI than their counterparts who were not employed. This agrees with studies [11,12,17,23] which reported that employed individuals tended to engage in premarital sex compared to those unemployed. Employed individuals have financial independence, personal freedom, social exposure, and access to information, possibly leading to more openness towards exploring sexual relations before marriage. Also, another plausible explanation for this finding could be that never-married women who were not working could easily be monitored by their parents than those who were working. We also observed that the richest never never-married women had lower odds of engaging in PSI than the poorest young women. This is consistent with the previous findings [2,3,7–9,11,12,17,23]. Social status and classification, less peer influence and substance use, and less exposure to risky sexual behaviour by the richest might have influenced the richest young women to less engage in PSI. As expected, we found that never-married women [15–24] who were exposed to mass media (television, radio, newspaper) had higher odds of engaging in PSI. This finding is similar to the results of studies [3,7–9,11,12,17,23–30] which reported that young women who were exposed to the media (radio, newspaper and television) had higher odds of engaging in PSI than those not exposed. This could be attributed to the effect of social media and exposure to pornography content on the sexual behaviour of those who were exposed to media to engage in PSI. At the contextual level, the study found higher odds of engaging in PSI among young women from the South-South and Southwest of the country compared to young women from the North Central. This is in line with studies [7–12,17,23–30] that reported that young women in some regions had higher odds of engaging in PSI than those from other regions. Gender roles across regions, sex and relationships, socio-cultural, religious practices, and attitudes towards procreation in some regions might have influenced never-married women in engaging PSI. The result further showed higher odds of engaging in PSI among young women from a community with medium knowledge of modern contraceptives and a community with high knowledge of modern contraceptives. This finding was similar to studies [11,12] which reported higher rates of PSI among communities with high knowledge of reproductive health and accessibility to contraception compared to those that lack knowledge. Though knowledge about contraception alone cannot prevent PSI, it could be attributed to high knowledge of how to prevent pregnancy. About the socio-economic status of the community, our finding revealed

that never-married women [15–24] with high community socio-economic status had higher odds of engaging in PSI compared to those with low community socio-economic status. This finding is similar to the results of previous studies [8,9] in which high odds of engaging in PSI were reported among communities with a high socio-economic status. A plausible reason for this finding could be attributed to social status and engagement in less risky sexual behaviour by people with high socio-economic status. Finally, the result showed that women in the 2018 survey year had lower odds of engaging in PSI than those young women in the 2008 survey year. Exposure to media and other social media and family planning information might have contributed to this relationship. Finally, the study revealed the level of disparities in PSI attributable to community-level factors rather than individual socio-demographic factors. This implies that PSI is influenced at multiple levels. Thus, interventions to reduce PSI may achieve more results if they are redeployed to give due attention to all levels of influence.

## Strengths and limitations

One major weakness of the DHS dataset is the fact that it is a cross-sectional survey thus, this study is limited because it cannot establish the causality of PSI. In addition to the weakness of the dataset is the fact that DHS information was collected through a self-reporting of retrospective events. There is the possibility of over-reporting or under-reporting of cases of PSI in this study. It should be noted that Nigeria is a highly religious country. Sensitive issues like PSI might have a tough traditional and religious tendency. Therefore, social desirability bias in self-reported sexual behaviour like PSI may lead to underreporting of stigmatized sexual behaviour and over-reporting of socially acceptable sexual behaviour. Thus, this study should be interpreted with caution based on the perceived social norms and culture in the country. Nevertheless, DHS datasets are high-quality population-based datasets. The pooling of the 15-year duration of three waves of NDHS datasets permits the analysis of the trends and patterns of change over ten years. In addition, the findings can be generalized and replicated. The findings could also be used to inform future research on PSI in Nigeria.

## Conclusions

The study concludes that even though a steady decrease in the prevalence of PSI among never-married women (15–24) over the 15-year duration of three waves of NDHS datasets (2008, 2013 and –2018) had been observed, it's crucial to assess the statistical significance of this trend in other further studies. Is the decline statistically significant, or could it be due to chance? Furthermore, the study has identified some predictors of PSI among never-married women (15–24). Factors such as age, education, religion, wealth status, media exposure, region of residence, community literacy level, knowledge of contraceptives, and socio-economic status were identified to be associated with PSI. Therefore, there is a need for policies and programmes that will include these predictors in promoting young women's reproductive and health rights in Nigeria. Thus, concerted efforts that will be well-tailored to achieve further drops in PSI should be welcomed to improve the general national health status of women.

## Author contributions

**Conceptualization:** Joseph Ayodeji Kupoluyi.

**Formal analysis:** Joseph Ayodeji Kupoluyi.

**Investigation:** Joseph Ayodeji Kupoluyi.

**Methodology:** Joseph Ayodeji Kupoluyi.

**Validation:** Joseph Ayodeji Kupoluyi.

**Writing – original draft:** Joseph Ayodeji Kupoluyi.

**Writing – review & editing:** Joseph Ayodeji Kupoluyi.

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
