## [Decision Letter · Decision Letter 0]

10 Oct 2024

PONE-D-24-33075Changes and predictors of premarital sex intercourse among never-married women (15-24 years) in Nigeria: A multilevel approach PLOS ONE

Dear Dr. Kupoluyi,

Thank you for submitting your manuscript to PLOS ONE. After careful consideration, we feel that it has merit but does not fully meet PLOS ONE’s publication criteria as it currently stands. Therefore, we invite you to submit a revised version of the manuscript that addresses the points raised during the review process.

We look forward to receiving your revised manuscript.

Kind regards,

Nega Degefa Megersa, Msc

Academic Editor

PLOS ONE

Journal Requirements:

Additional Editor Comments:

Abstract

The abstract is well-written and concisely summarized.

Limit the abbreviations in the abstract section. Thus, put IR, ICC, and PCV in full on their initial appearance.

The prevalence of PSI over ten years was described as 31.8%, consider 95% CI to provide a more complete picture of the problem by quantifying uncertainty, aiding in comparisons, informing decision-making, and evaluating the reliability of estimates.

Introduction

Thus, this study could inform the development of policy on premarital sexual intercourse among young women in Nigeria.

How this study could inform policy and what areas should the policy address to further hasten the declining PSI rate in Nigeria?

Method

Missing values on age at first sex are considered as not having had sexual intercourse and are included in both the numerator and denominator.

What does this mean, would you please explain it? What techniques have you employed to manage your missing values?

Conclusion

The conclusion appears to be well-supported by the study's data. However, a more comprehensive analysis, addressing the points mentioned below, would enhance the conclusion.

While a decrease in prevalence is observed, it's crucial to assess the statistical significance of this trend. Is the decline statistically significant, or could it be due to chance?

Acknowledging the limitations of the research, strengthens the credibility of the conclusion.

It is beneficial if specific predictors of premarital sexual intercourse are included instead of referring to predictors of PSI in general.

It could be strengthened by explicitly outlining specific policy recommendations based on the study's findings, which would make the research more doable.

Reviewers' comments:

Reviewer's Responses to Questions

**Comments to the Author**

1. Is the manuscript technically sound, and do the data support the conclusions?

Reviewer #1: Partly

Reviewer #2: Partly

2. Has the statistical analysis been performed appropriately and rigorously? 

Reviewer #1: Yes

Reviewer #2: Yes

3. Have the authors made all data underlying the findings in their manuscript fully available?

Reviewer #1: Yes

Reviewer #2: Yes

4. Is the manuscript presented in an intelligible fashion and written in standard English?

Reviewer #1: Yes

Reviewer #2: Yes

5. Review Comments to the Author

Reviewer #1: After reviewing the manuscript titled "Changes and predictors of premarital sex intercourse among never-married women (15-24 years) in Nigeria: A multilevel approach," here are some key areas of improvement.

Research Gap:

The introduction outlines the consequences of premarital sexual intercourse (PSI) among young women, but it does not sufficiently highlight the novel contribution of this study. While the authors mentioned that existing studies focus on adolescents, there is a need to emphasize the lack of longitudinal data or multilevel approaches in previous research. The text should explicitly state how this study fills the gap by using 10-year pooled data and a multilevel binary logistic regression model to track PSI trends over time.

Suggested Improvements:

• Line 56-61: Strengthen the argument for why a multilevel approach is necessary for understanding PSI and why existing single-level analyses are insufficient. Discuss why previous studies lacked this dimension and how the author's study adds value.

Specific Lines for Improvement:

• Line 28-30: Clarify the unique contribution of the author's work in comparison to previous studies, such as by stating, “This study is the first to use a multilevel analysis approach across a 10-year dataset to investigate changes in PSI and its predictors.”

• Line 72-75: Expand on the limitations of prior research by clearly stating the gaps in the context of multilevel modeling and the lack of longitudinal studies, especially in Nigeria.

• Line 102: Further justification is needed for using this particular dataset and why the pooled 10-year NDHS data offers new insights that previous studies could not.

Grammatical Issues and Improvements:

• Line 25: "a multilevel approach" – should be rephrased to "using a multilevel approach" for clarity.

• Line 78: "Prevalence of PSI decreased steadily" – consider rephrasing to "The prevalence of PSI steadily decreased" to improve readability.

• Line 120-122: "The ICC value of 10.1% and the PCV of 41.3% show that the individual and community factors were important in explaining the variations in PSI." – This sentence can be made more concise and clear. Suggested: "An ICC value of 10.1% and a PCV of 41.3% indicate that both individual and community factors significantly influenced PSI variations."

• Line 180: "empower young women to achieve a further drop in PSI" – should be revised to "empower young women to contribute to a further reduction in PSI" for a more academic tone.

Methodological Issues and Suggestions for Improvement:

1. Line 32-35: The rationale for pooling data from the 2008, 2013, and 2018 surveys is briefly mentioned. However, it is crucial to provide more justification for why pooling these datasets is appropriate for studying changes in PSI over time. A clearer explanation of how trends in these periods will inform the research objectives would strengthen the methodology.

2. Line 67-70: The study design description should expand on how cross-sectional data limits the ability to infer causality. Cross-sectional studies are limited in their ability to determine temporal relationships, which should be explicitly acknowledged in the limitations.

3. Line 80-85: The method used for handling missing data is not well elaborated. It would benefit the manuscript to detail if any imputation methods were applied to handle missing values or whether complete case analysis was used.

4. Line 90-92: The explanation of multilevel modeling is good, but the manuscript would benefit from including a brief description of why this modeling technique was chosen, particularly regarding the hierarchical structure of the data (i.e., individual and community levels).

5. Line 102: It is important to elaborate on the rationale behind selecting the specific contextual variables for inclusion in the model. While it is mentioned that they are based on previous literature, more detail is needed on how these factors were prioritized and why others may have been excluded.

6. Line 112-115: The results of the multicollinearity test (variance inflation factor, VIF) are mentioned but not fully explained. The manuscript should include what threshold of VIF was used to determine multicollinearity and how any problematic variables were addressed.

Reviewer #2: Changes and predictors of premarital sex intercourse among never-married women (15-24 years) in Nigeria: A multilevel approach

The author has addressed an important topic of social and public health interest. The prevalence of PSI globally has been identified to be high and could result in unwanted pregnancy, school dropout among females, sexually transmitted diseases, etc. But I have the following observations and comments, which the author may wish to consider:

Abstract

You have stated that the ICC of 10.1% and PCV of 48.3% imply that individual and community factors are important. Do you mean the ICC explained the importance of individual factors, while the PCV explains the importance of community factors? This may be confusing to some readers.

Background study

1. The topic covers the changes and the predictors of PSI, but you have discussed what literature found on changes and prevalence without reference to what previous literature found on the predictors.

2. There were no explicit stated aims and objectives in the study.

Methods

Data source

1. You have stated you used a 10-year pooled data set. Do you have any specific reasons for choosing 10 years out of the existing six waves that have spanned over 30 years?

2. The three data sets actually spanned over 15 years of the duration, with 3 waves. I am not sure they are '10-year waves' as stated. For instance, the 2008 wave covered 2003-2008, 2013 covered 2008-2013, and 2018 covered 2013-2018. This is a 15-year duration of 3 waves (2008, 2013, 2018).

Outcome variables

1. What does ‘never unmarried’ mean? This may be a typo-error. Check it

2. How did you handle the missing data?

Predictors

1. I suppose that the unit of analysis from this study is ‘never-married’ women aged 15 to 24 years. The outcome variable, PSI, is an onset at a point in time. I am wondering how some individual characteristics that came to be after the onset of the outcome can be considered a predictor or a determinant of the status (PSI). For instance, a partner's education status, parity, child mortality, etc. Moreover, some of these predictors were listed but were missing in the analysis.

Statistical Analyses

1. I am wondering if STATA 14 could handle some features in mixed effect analysis. Many updates have occurred since Stata 14 was released. Can you trust the results you got? In addition, I am not sure if Stata 14 could handle some features in NDHS 2018. Please verify this.

2. Rewrite the ICC and PCV formulas in equation format if you need to keep them in the paper.

Handling missing values statement

1. What do you mean by ‘numerator’ and 'denominator’?

2. Was age the only variable with missing values?

3. How were the missing values in other variables handled? In other words, what was the magnitude of missingness in the data?

Results

1. Table 2: Why was the chi-sq for age in 2008 having a triple star (***)?

Fixed effects

The reporting style here is confusing. You applied multilevel analysis (having fixed effect and random effect components). Under the ‘fixed effect’ heading, you reported the random effect, while almost nothing was reported under ‘random effect.’

Discussion

This statement lowers the odds of PSI among Islamic women over other religions. You said the reason is that ‘the teaching and practices of Islam frown upon PSI’. This is controversial because other religions (i.e., Christianity) also teach and frown at PSI. Do you have references to support your claims?

Some of the predictors of PSI listed in the paper were not used in the analysis and were not discussed. For instance, ‘partner’s education' and ‘ever experience child mortality’ were listed. However, 'sex of household head’ could be a confounding variable but was not included in the analysis.

Conclusion

1. Too brief

2. What were the implications of the findings, and could they be used to improve the PSI situation in Nigeria?

3. Make a statement on the direction of future study.

6. PLOS authors have the option to publish the peer review history of their article (what does this mean? ). If published, this will include your full peer review and any attached files.

**Do you want your identity to be public for this peer review?** For information about this choice, including consent withdrawal, please see our Privacy Policy .

Reviewer #1: **Yes: ** Obasanjo Bolarinwa

Reviewer #2: **Yes: ** Phillips Edomwonyi Obasohan

---

## [Author Response · Author response to Decision Letter 1]

30 Oct 2024

Journal Requirements:

Comment 1: Please ensure that your manuscript meets PLOS ONE's style requirements, including those for file naming. The PLOS ONE style templates can be found at

Response: Thank you. The PLOS ONE’s style requirements (i.e. Title, Author, Affiliations Formatting Guidelines and Manuscript Body Formatting Guidelines) have been strictly followed.

Comment 2: PLOS requires an ORCID iD for the corresponding author in Editorial Manager on papers submitted after December 6th, 2016. Please ensure that you have an ORCID iD and that it is validated in Editorial Manager. To do this, go to ‘Update my Information’ (in the upper left-hand corner of the main menu), and click on the Fetch/Validate link next to the ORCID field. This will take you to the ORCID site and allow you to create a new iD or authenticate a pre-existing iD in Editorial Manager.

Response: Thank you, ORCID ID (http://orcid.org/0000-0002-5645-7689) has been validated in Editorial Manager.

Comment 3: Your ethics statement should only appear in the Methods section of your manuscript. If your ethics statement is written in any section besides the Methods, please delete it from any other section.

Response: Thank you. As suggested, the Ethics statement that appears in any section besides the methods Section has been deleted. Ethics statements now appear only in the Methods section of the Manuscript.

Additional Editor Comments:

Abstract

Comment 1: The abstract is well-written and concisely summarized. Limit the abbreviations in the abstract section. Thus, put IR, ICC, and PCV in full on their initial appearance.

Response: Thank you. As suggested, the abbreviations in the abstract have been corrected. IR has been replaced with Individual recode, ICC has been replaced with Intra-Class Correlation, and PCV has been replaced with Proportional Change in Variance.

Comment 2: The prevalence of PSI over ten years was described as 31.8%, consider 95% CI to provide a more complete picture of the problem by quantifying uncertainty, aiding in comparisons, informing decision-making, and evaluating the reliability of estimates.

Response: Thank you. Agreed. As suggested 95%CI has been included.

Introduction

Comment: Thus, this study could inform the development of policy on premarital sexual intercourse among young women in Nigeria. How this study could inform policy and what areas should the policy address to further hasten the declining PSI rate in Nigeria?

Response: Thank you for your suggestion. An area to further hasten the declining PSI rate in Nigeria has been added.

Method

Comment: Missing values on age at first sex are considered as not having had sexual intercourse and are included in both the numerator and denominator. What does this mean, would you please explain it? What techniques have you employed to manage your missing values?

Response: Thank you. In the DHS, a missing value means that the variable is not applicable or there was not a valid response. Thus, we handled the missing data by declaring them as missing. Thus, dropping/removing the values/data/observation from the analysis.

Conclusion

Comment 1: The conclusion appears to be well-supported by the study's data. However, a more comprehensive analysis, addressing the points mentioned below, would enhance the conclusion. While a decrease in prevalence is observed, it's crucial to assess the statistical significance of this trend. Is the decline statistically significant, or could it be due to chance?

Response: Thank you for your suggestion. The following statement has been added to read: ‘While a decrease in prevalence is observed, it's crucial to assess the statistical significance of this trend. Is the decline statistically significant, or could it be due to chance?)’

Comment 2: Acknowledging the limitations of the research, strengthens the credibility of the conclusion. It is beneficial if specific predictors of premarital sexual intercourse are included instead of referring to predictors of PSI in general. It could be strengthened by explicitly outlining specific policy recommendations based on the study's findings, which would make the research more doable.

Response: Thank you for providing this insight. Predictors of premarital sexual intercourse have been included.

Comments to the Author

Reviewer #1: Obasanjo Bolarinwa

Research Gap:

Comment: The introduction outlines the consequences of premarital sexual intercourse (PSI) among young women, but it does not sufficiently highlight the novel contribution of this study. While the authors mentioned that existing studies focus on adolescents, there is a need to emphasize the lack of longitudinal data or multilevel approaches in previous research. The text should explicitly state how this study fills the gap by using 10-year pooled data and a multilevel binary logistic regression model to track PSI trends over time.

Response: Thank you. It has been added to read:

‘Most of these studies on PSI have documented a range of individual-level factors such as; respondent’s age, region of residence, place of residence, wealth status, employment status, religion, and education among others as predictors of PSI. To the best of our knowledge, there is a paucity of information on studies on community-level factors on PSI in Nigeria. In addition to this, there is a dearth of studies on the general change in premarital sexual intercourse among young women using longitudinal nationally representative data, to examine the progress made in reducing PSI in Nigeria. Thus, this study will fill this knowledge gap by using pooled nationally representative data and a multilevel binary logistic regression model to track PSI trends and their predictors over time in Nigeria. The use of 15-year duration of 3- waves NDHS data offers new insights that previous studies could not. Thus, this study aimed to investigate changes and predictors of premarital sexual intercourse among young women. Assessing the changes and predictors of PSI could inform the development of policies and programmes on PSI among young women in Nigeria that will empower young women to make informed choices about their sexual and reproductive health rights.’

Suggested Improvements:

Comment: • Line 56-61:

Strengthen the argument for why a multilevel approach is necessary for understanding PSI and why existing single-level analyses are insufficient. Discuss why previous studies lacked this dimension and how the author's study adds value.

Response: Thank you. It has been added as suggested (See response to Research Gap comment 1 above)

Specific Lines for Improvement:

Comment: Line 28-30:

Clarify the unique contribution of the author's work in comparison to previous studies, such as by stating, “This study is the first to use a multilevel analysis approach across a 10-year dataset to investigate changes in PSI and its predictors.”

Response: Thank you. It has been added (See response to Research Gap comment 1 above)

Comment: Line 72-75: Expand on the limitations of prior research by clearly stating the gaps in the context of multilevel modeling and the lack of longitudinal studies, especially in Nigeria.

Response: Thank you. Limitations of prior research in multilevel and longitudinal studies have been expanded as suggested (See response to Research Gap comment 1)

Comment: Line 102: Further justification is needed for using this particular dataset and why the pooled 10-year NDHS data offers new insights that previous studies could not.

Response: Thank you. Justification for using a 15-year duration of 3 waves NDHS has been added (See response to Research Gap above).

Grammatical Issues and Improvements:

Comment: Line 25: "a multilevel approach" – should be rephrased to "using a multilevel approach" for clarity.

Response: Thank you for your suggestion. It has been corrected to read "using a multilevel approach"

Comment: Line 78: "Prevalence of PSI decreased steadily" – consider rephrasing to "The prevalence of PSI steadily decreased" to improve readability.

Response: Thank you for your suggestion. It has been corrected to read "The prevalence of PSI steadily decreased"

Comment: Line 120-122: "The ICC value of 10.1% and the PCV of 41.3% show that the individual and community factors were important in explaining the variations in PSI." – This sentence can be made more concise and clear. Suggested: "An ICC value of 10.1% and a PCV of 41.3% indicate that both individual and community factors significantly influenced PSI variations."

Response: Thank you for your suggestion. It has been corrected as suggested.

Comment: Line 180: "empower young women to achieve a further drop in PSI" – should be revised to "empower young women to contribute to a further reduction in PSI" for a more academic tone.

Response: Thank you for your suggestion. It has been corrected to read: ‘to empower young women to contribute to a further reduction in PSI to improve the general national health status of women and to ensure progress towards achieving a reduction in early pregnancies among unmarried women’

Methodological Issues and Suggestions for Improvement:

Comment: Line 32-35: The rationale for pooling data from the 2008, 2013, and 2018 surveys is briefly mentioned. However, it is crucial to provide more justification for why pooling these datasets is appropriate for studying changes in PSI over time. A clearer explanation of how trends in these periods will inform the research objectives would strengthen the methodology.

Response: Thank you. That is an interesting query. It has been added to read. ‘The datasets were pooled to increase the sample size and enable studying changes in PSI over time’.

Comment: Line 67-70: The study design description should expand on how cross-sectional data limits the ability to infer causality. Cross-sectional studies are limited in their ability to determine temporal relationships, which should be explicitly acknowledged in the limitations.

Response: Thank you. The Limitation of a cross-sectional study design has been acknowledged and incorporated.

Comment: Line 80-85: The method used for handling missing data is not well elaborated. It would benefit the manuscript to detail if any imputation methods were applied to handle missing values or whether complete case analysis was used.

Response: Thank you. Missing data were dropped. A complete case was deleted in the data analysis.

Comment: Line 90-92: The explanation of multilevel modeling is good, but the manuscript would benefit from including a brief description of why this modeling technique was chosen, particularly regarding the hierarchical structure of the data (i.e., individual and community levels).

Response: Thank you. It has been added to read: ‘Finally, owing to the hierarchical nature of the NDHS data and the objective of this study, a four-model multi-level binary logistic regression was constructed to examine the individual and community level factors associated with PSI’.

Comment: Line 102: It is important to elaborate on the rationale behind selecting the specific contextual variables for inclusion in the model. While it is mentioned that they are based on previous literature, more detail is needed on how these factors were prioritized and why others may have been excluded.

Response: Thank you. It has been added to read:

Odds ratio (OR) and 95% confidence intervals (CI) were provided for all models in the multivariable analysis while multicollinearity was done using variance inflation factor (VIF) to exclude explanatory variables with evidence of collinearity.

Comment: Line 112-115: The results of the multicollinearity test (variance inflation factor, VIF) are mentioned but not fully explained. The manuscript should include what threshold of VIF was used to determine multicollinearity and how any problematic variables were addressed.

Response: Thank you for providing these insights. It has been added to read: ‘A Multicollinearity test was performed using variance inflation factor (VIF < 5). All explanatory variables with evidence of no collinearity (mean VIF= 1.38, maximum = 1.87 and minimum VIF = 1.10) were retained in the models while those with evidence of collinearity (VIF > 5) were excluded from the models.’

Reviewer #2: Phillips Edomwonyi Obasohan

Changes and predictors of premarital sex intercourse among never-married women (15-24 years) in Nigeria: A multilevel approach

The author has addressed an important topic of social and public health interest. The prevalence of PSI globally has been identified to be high and could result in unwanted pregnancy, school dropout among females, sexually transmitted diseases, etc. But I have the following observations and comments, which the author may wish to consider:

Abstract

Comment: You have stated that the ICC of 10.1% and PCV of 48.3% imply that individual and community factors are important. Do you mean the ICC explained the importance of individual factors, while the PCV explains the importance of community factors? This may be confusing to some readers.

Response: Thank you. That is an interesting query. We have incorporated your comments to read: The Intraclass correlation coefficient (ICC) explained the proportion of total variance in the PSI that is attributable to the area level while the proportional change in variance (PCV) explained the variability in the odds of PSI explained by each of the models Therefore, ICC of 10.0% shows that the individual and community factors were important in explaining the variations in PSI between clusters. The PCV of 41.3% explained the variability in the odds of PSI explained by each of the models with more terms.

Background study

Comment 1: The topic covers the changes and the predictors of PSI, but you have discussed what literature found on changes and prevalence without reference to what previous literature found on the predictors.

Response: Thank you for this observation. As suggested, it has been added to read: ‘Most of these studies on PSI have documented a range of individual-level factors such as; respondent’s age, region of residence, place of residence, wealth status, employment status, religion, and education among others as predictors of PSI’.

Comment 2: There were no explicit stated aims and objectives in the study.

Response: Thank you. We have elaborated on it to read: ‘Thus, this study aimed at investigating changes and predictors of premarital sexual intercourse among young women’

Methods

Data source

Comment 1: You have stated you used a 10-year pooled data set. Do you have any specific reasons for choosing 10 years out of the existing six waves that have spanned over 30 years?

Response: Thank you for noting this. The study used the last three waves of NDHS to map out major trends in the country focusing on young women and to evaluate the extent of various national variations in PSI. Pooling data provides an opportunity to increase the number of observations, strength of evidence, and analyse of a much larger data set. It makes it possible to access diverse data and perform analysis on data that is updated regularly. It enhanced statistical power, the ability to compare outcomes and validate models across sites or settings, and opportunities to develop new measures. The large size of the data enhances the quality of the original studies that were combined, the statistical methods used for the pooled analysis. It increased study power that permits a full examination of effect modification within the data.

Comment 1: The three data sets actually spanned over 15 years of the duration, with 3 waves. I am not sure they are '10-year waves' as stated. For instance, the 2008 wave covered 2003-2008, 2013 covered 2008-2013, and 2018 covered 2013-2018. This is a 15-year duration of 3 waves (2008, 2013, and 2018).

Respon

---

## [Decision Letter · Decision Letter 1]

22 Jan 2025

PONE-D-24-33075R1Changes and predictors of premarital sex intercourse among never-married women (15-24 years) in Nigeria: A multilevel approachPLOS ONE

Dear Dr. Kupoluyi,

Thank you for submitting your manuscript to PLOS ONE. After careful consideration, we feel that it has merit but does not fully meet PLOS ONE’s publication criteria as it currently stands. Therefore, we invite you to submit a revised version of the manuscript that addresses the points raised during the review process.

We look forward to receiving your revised manuscript.

Kind regards,

Omid Dadras, MD, PhD

Academic Editor

PLOS ONE

Journal Requirements:

Reviewers' comments:

Reviewer's Responses to Questions

**Comments to the Author**

1. If the authors have adequately addressed your comments raised in a previous round of review and you feel that this manuscript is now acceptable for publication, you may indicate that here to bypass the “Comments to the Author” section, enter your conflict of interest statement in the “Confidential to Editor” section, and submit your "Accept" recommendation.

Reviewer #3: (No Response)

2. Is the manuscript technically sound, and do the data support the conclusions?

Reviewer #3: (No Response)

3. Has the statistical analysis been performed appropriately and rigorously? 

Reviewer #3: (No Response)

4. Have the authors made all data underlying the findings in their manuscript fully available?

Reviewer #3: (No Response)

5. Is the manuscript presented in an intelligible fashion and written in standard English?

Reviewer #3: (No Response)

6. Review Comments to the Author

Reviewer #3: Peer Review Report

Title: "Changes and predictors of premarital sex intercourse among never-married women (15-24 years) in Nigeria: using a multilevel approach"

General Assessment:

This manuscript presents a comprehensive analysis of premarital sexual intercourse (PSI) trends and determinants among young women in Nigeria using nationally representative data across three time points. The study makes a valuable contribution to understanding both individual and community-level factors influencing PSI, with important implications for public health interventions and policy.

Strengths:

The study demonstrates several notable strengths through its methodological approach and analytical depth. The use of nationally representative data across three time points (2008-2018) allows for robust temporal trend analysis. The multilevel analytical approach appropriately accounts for the hierarchical nature of the data and enables examination of both individual and community-level effects. The theoretical framework is well-conceived and appropriately applied.

Major Concerns:

Methodological Issues:

The handling of missing data requires more detailed explanation. While the authors mention excluding missing values, a more thorough discussion of the potential impact of these exclusions on the results is needed. The potential for social desirability bias in self-reported sexual behavior data should be more thoroughly addressed in the limitations section.

Analytical Concerns:

The statistical significance of the observed temporal trends in PSI requires more rigorous testing. While the author note a declining trend, formal tests for trend should be included. The interaction effects between individual and community-level factors warrant exploration.

Results Presentation:

The presentation of results could be improved through clearer organization of the findings section, more consistent reporting of confidence intervals, and more detailed interpretation of the random effects results.

Specific Recommendations:

Introduction:

The rationale for the multilevel approach needs strengthening, along with more context about Nigeria's sexual and reproductive health policies. The theoretical framework should be better integrated with existing literature.

Methods:

The author should provide more detail about the sampling strategy and power calculations. The construction of community-level variables needs clarification. Sensitivity analyses for key assumptions should be included. The choice of variables included in each model requires better justification.

Results:

Formal tests for temporal trends should be added to support the observed patterns. Interaction analyses between key variables would strengthen the findings. The random effects require more detailed interpretation. The addition of predicted probabilities for key findings would enhance understanding of the results.

Discussion:

The comparison with other studies from similar contexts needs strengthening. Policy implications of the findings should be expanded. The sustainability of observed improvements requires more thorough examination. The implications of the community-level effects warrant deeper discussion.

The manuscript makes a valuable contribution to understanding PSI patterns and determinants in Nigeria. With appropriate revisions addressing the above concerns, it would make a strong contribution to the literature.

7. PLOS authors have the option to publish the peer review history of their article (what does this mean? ). If published, this will include your full peer review and any attached files.

**Do you want your identity to be public for this peer review?** For information about this choice, including consent withdrawal, please see our Privacy Policy .

Reviewer #3: No

---

## [Author Response · Author response to Decision Letter 2]

25 Feb 2025

Major Concerns:

1. Methodological Issues:

Query 1: The handling of missing data requires more detailed explanation. While the authors mention excluding missing values, a more thorough discussion of the potential impact of these exclusions on the results is needed.

Answer: Thank you for this observation. The percentage of missing data is low (less than 1). Therefore, missing data has no practical effect on the analysis. Also, to ensure that statistical estimates based on DHS data are valid, as recommended by DHS, sample weights and STATA survey command (svy) command in the analysis were applied to adjust for the over/under-sampling and non-response during the survey.

Query 2: The potential for social desirability bias in self-reported sexual behaviour data should be more thoroughly addressed in the limitations section.

Answer: Thank you for this observation. The potential for social desirability bias in self-reported sexual behaviour data has been addressed in the limitations section. It now read: ‘In addition to the weakness of the dataset is the fact that DHS information on sexual behaviour was collected through a self-reporting of retrospective events. Thus, there is the possibility of over-reporting or under-reporting of cases of PSI in this study. Therefore, the study should be interpreted with caution based on the perceived social norms and culture.

2. Analytical Concerns:

Query 1: The statistical significance of the observed temporal trends in PSI requires more rigorous testing. While the author note a declining trend, formal tests for trend should be included.

Answer: Thank you for this observation. Trend detention in time was conducted using Mann Kendall test. It is a test that helps us know whether a trend exists in a time and whether it is upward or downward and statistically significant. In this study, the trend is monotonic downward and significant (p < 0.001). This has been specified in the manuscript.

Query 2: The interaction effects between individual and community-level factors warrant exploration.

Answer: Thank you. Interaction effect analyses between individual and community-level factors implies another level of analysis which future study could explore. However, this has been explained further.

3. Results Presentation:

Query 1: The presentation of results could be improved through clearer organization of the findings section, more consistent reporting of confidence intervals, and more detailed interpretation of the random effects results.

Answer: Thank you. The result section has been improved. All confidence intervals has been reported. Also, more detailed interpretation of the random effects results has been done.

4. Specific Recommendations:

Introduction:

Ouery 1: The rationale for the multilevel approach needs strengthening, along with more context about Nigeria's sexual and reproductive health policies.

Answer: Thank you for this observation. The rationale for multi-level has been strengthen to read:

‘Community-level factors refer to different characteristics of the social group to which individual belong. These characteristics may affect PSI either directly or indirectly by controlling how individuals’ characteristics affect PSI. It is therefore important to know the effect of community-level variables on individual–level and household-level outcomes on PSI. The availability of DHS hierarchical data in which individuals are clustered within household and household clustered within the communities, give rooms to account for the effect of both the individual and group-level influence on PSI in Nigeria’.

Furthermore, more context about Nigeria’s sexual and reproductive health policies have been included. It reads:

‘The national sexual and reproductive health policies has been anchored within the framework of the national health policy. The policy recognizes the implementation of reproductive health within the context .of primary health care, with the goal of ensuring availability and access to complete sexual and reproductive health information and quality of services. The policy largely discouraged PSI with a strongly but not openly promoting abstinence until marriage as ideal behaviour. Under the policy, PSI among young women aged 15-24 is considered illegal. However, the policy provides a comprehensive sex education and contraceptive services to young people to help them make informed decisions about sexual health, including preventing unwanted pregnancies and sexually transmitted infections (STIs). In addition, the policy accentuates the need for access to reproductive health services for those who are sexually active but do not aggressively criminalise premarital sex. The social and cultural norms around the policy vary depending on region and religious beliefs. Thus, making some of these policies and programmes largely ineffective’.

Query 2: The theoretical framework should be better integrated with existing literature.

Answer: Thank you. The theoretical framework has been included to read:

‘This study theoretical framework was drawn from the Theory of Planned Behaviour and Social Cognitive Theory. The theories emphasized on the role of individual attitudes in considering whether PSI is morally acceptable or not; feels of engaging in PSI or not (social norms), has access to contraception and personal confidence (perceived control), and has personal beliefs to influencing decisions about PSI, while considering factors such as family background, peer pressure, religious values, and personal commitment to a partner’.

5. Methods:

Query 1: The author should provide more detail about the sampling strategy and power calculations.

Answer: Thank you. More details about the sampling strategy and power calculations has been added. It now read: ‘DHS used a two-stage sampling procedure to collect data from all the 36 states and the federal capital territory (FCT), Abuja, Nigeria. The samples for the survey were drawn randomly from cluster or enumeration areas (EAs). The EAs served as the primary sampling unit for the survey. A systematic selection of households from the list of households were done to interview eligible respondents. Women aged 15-49 and men aged 15-64 were selected systematically from the households and interviewed’.

Query 2: The construction of community-level variables needs clarification.

Answer: Thank you. The construction of all the community level variables has been added to read:

‘--- a combined frequencies of reading newspaper, watching television and listening to radio within a week. The scores were further distributed into two groups (exposed and not exposed). DHS measured wealth index as a uniform composite variable determined through principal component analysis (PCA) and was built on household properties such as: water supply, television, electricity, radio, refrigerator, type of flooring, and type of vehicle. The scores were categorised into five quintiles, and each quintile represents a relative measure of a household’s socioeconomic status. Contextual variables considered in this study were community socioeconomic status, community knowledge of modern contraceptives, community literacy and region of residence. The region of residence was measured and divided into five geopolitical groups namely: South West, South East, North West, North East, North Central, and South-South. Other contextual variables were constructed and classified by aggregating individual-level factors at the cluster level using the median value as a cut-off point. These variables were: community socioeconomic status (low, moderate, high), community literacy (low, moderate, high), and community knowledge of modern contraceptives (low, moderate, high). The community socioeconomic status was generated from the combination of individual wealth quintile while community literacy was generated from the combination of individual literacy level. Community knowledge of modern contraceptives was generated from the combination of individual knowledge of any contraceptive methods’.

Query 3: Sensitivity analyses for key assumptions should be included.

Answer: Thank you for this observation. A Mann Kendall test was performed. The distribution with other variable show a fair distribution.

Query 4: The choice of variables included in each model requires better justification.

Answer: Thank you. It has been stated earlier in the manuscript (see explanatory variables section, line 1-4) that ‘All the explanatory variables used for this study were selected based on the established association reported in the previous studies with PSI and their level of significance (p < 0.05)’. A Multicollinearity test was performed using a variance inflation factor (VIF < 5). All explanatory variables with evidence of no collinearity (mean VIF= 1.38, maximum = 1.87 and minimum VIF = 1.10) were retained in the models while those with evidence of collinearity (VIF > 5) were excluded from the models.

6. Results:

Query 1: Formal tests for temporal trends should be added to support the observed patterns.

Answer: Thank you for this observation. Trend detention in time has been conducted using Mann Kendall test. The result shows that the data series has a statistically significant trend at p< 0.001

Query 2: Interaction analyses between key variables would strengthen the findings.

Answer: Thank you. Interaction analyses between key variables implies another level of analysis which further study could explore.

Query 3: The random effects require more detailed interpretation.

Answer: Thank you for this observation. More explanations of the random effects has been done.

Ouery 4: The addition of predicted probabilities for key findings would enhance understanding of the results.

Answer: Thank you for this observation. Predicted probabilities for the key findings have been added.

7. Discussion:

Query 1: The comparison with other studies from similar contexts needs strengthening.

Answer: Thank you for the observations. The comparison with other studies has been done to strengthen the manuscript.

Query 2: Policy implications of the findings should be expanded.

Answer: Thank you. Policy implications of the findings has been expanded.

Query 3: The sustainability of observed improvements requires more thorough examination.

Answer: Thank you. The sustainability of observed improvements has been examined and expanded.

Query 4: The implications of the community-level effects warrant deeper discussion.

Answer: Thank you. The implications of the community-level effects has been discussed.

---

## [Editor Report · Decision Letter 2]

11 Mar 2025

Changes and predictors of premarital sex intercourse among never-married women (15-24 years) in Nigeria: a multilevel approach

PONE-D-24-33075R2

Dear Dr. Joseph Ayodeji Kupoluyi,

We’re pleased to inform you that your manuscript has been judged scientifically suitable for publication and will be formally accepted for publication once it meets all outstanding technical requirements.

Kind regards,

Omid Dadras, MD, PhD

Academic Editor

PLOS ONE
---

## [Editor Report · Acceptance letter]

PONE-D-24-33075R2

PLOS ONE

Dear Dr. Kupoluyi,

I'm pleased to inform you that your manuscript has been deemed suitable for publication in PLOS ONE. Congratulations! Your manuscript is now being handed over to our production team.

Kind regards,

on behalf of

Dr Omid Dadras

Academic Editor

PLOS ONE